# Universal Glia to Neurone Lactate Transfer in the Nervous System: Physiological Functions and Pathological Consequences

**DOI:** 10.3390/bios10110183

**Published:** 2020-11-19

**Authors:** Carolyn L. Powell, Anna R. Davidson, Angus M. Brown

**Affiliations:** 1School of Life Sciences, University of Nottingham, Nottingham NG7 2UH, UK; mbyclp@exmail.nottingham.ac.uk (C.L.P.); mbyard@nottingham.ac.uk (A.R.D.); 2Department of Neurology, University of Washington, Seattle, WA 98105, USA

**Keywords:** lactate, glucose, glycogen, astrocyte, neurone, monocarboxylate transporter, lactate dehydrogenase

## Abstract

Whilst it is universally accepted that the energy support of the brain is glucose, the form in which the glucose is taken up by neurones is the topic of intense debate. In the last few decades, the concept of lactate shuttling between glial elements and neural elements has emerged in which the glial cells glycolytically metabolise glucose/glycogen to lactate, which is shuttled to the neural elements via the extracellular fluid. The process occurs during periods of compromised glucose availability where glycogen stored in astrocytes provides lactate to the neurones, and is an integral part of the formation of learning and memory where the energy intensive process of learning requires neuronal lactate uptake provided by astrocytes. More recently sleep, myelination and motor end plate integrity have been shown to involve lactate shuttling. The sequential aspect of lactate production in the astrocyte followed by transport to the neurones is vulnerable to interruption and it is reported that such disparate pathological conditions as Alzheimer’s disease, amyotrophic lateral sclerosis, depression and schizophrenia show disrupted lactate signalling between glial cells and neurones.

## 1. Introduction

The brain has an absolute requirement for glucose as its main source of energy substrate [1]. Ingested carbohydrate is metabolised to glucose in the gastro-intestinal tract and then travels to the liver via the hepatic portal vein, where excess glucose is stored as glycogen [2]. During periods of falling systemic glucose, the liver glycogen is metabolised to glucose, which is released into the systemic circulation to fuel brain activity [3]. In this manner, it can be unequivocally stated that the role of liver glucose is to fuel the brain via increasing glucose supply to the brain via the systemic circulation [1], and although all other organs in the body have access to this glucose via their blood supply, the brain is the most sensitive to deficits in blood glucose delivery. When deficits occur in systemic glucose concentration such that brain energy demand exceeds supply, brain function is compromised and loss of consciousness, seizure and coma can occur [3]. The respiratory coefficient (RC) is the ratio of carbon dioxide released versus the oxygen consumed [1]. In the human brain, this value approaches one, evidence that carbohydrates are the prime brain fuel. The fate of glucose in the brain has emerged as one of the great mysteries in neuroscience, a controversial topic that has polarised opinion. Given the extremely high metabolic rate of the brain when compared with other organs, it is of prime importance that blood glucose is delivered in excess of demand with a healthy safety factor to ensure brief transient mismatches between demand and supply do not occur [4]. The accepted dogma was that all brain cells, neurones and glia, directly take up glucose via glucose transporters expressed on their membrane, where the glucose is metabolised intracellularly, the first step being glycolytic metabolism of glucose to pyruvate/lactate, followed by oxidative metabolism, where pyruvate, subsequently converted to acetyl CoA, is taken up into the mitochondria and metabolised to produce carbon dioxide and water plus ATP [2]. Under such a scheme, the metabolic rates of glucose uptake versus oxygen consumption would be six, since
C_6_H_12_O_6_ + 6O_2_ ⇒ 6CO_2_ + 6H_2_O

However, the advent of a technique that allowed measurement of radio-labelled de-oxyglucose found a ratio of the cerebral metabolic rates (CMR) of between 5.5 and 5.8 at rest, which indicated incomplete oxidation of glucose, i.e., not all of the pyruvate/lactate that was produced was fed into the oxidative pathway [5]. This anomalous finding caused much interest and instigated more detailed analysis of brain metabolism, which continues unabated to this day. The reason that the issue has not been satisfactorily resolved can be explained by (i) limited resolution of available techniques, which means that the metabolic rate of single neurones in vivo cannot be measured; (ii) there are obvious differences seen between the CMR of glia versus neurones, but given their close apposition it is difficult to resolve any differences in situ; (iii) cells display a versatility in their metabolism that is dependent upon the state of activity and the availability of substrates that confuses even the most sensitive of measurements in vivo; and (iv) given the extremely narrow extracellular space, it is impossible to measure localised concentrations of substrate and metabolites in real time in response to metabolic challenges or changes in activity [6]. This frustrating situation ensures that modelling involving multi-compartment simulations are in widespread use [7,8]. The experimental manoeuvres associated with such modelling studies must be carefully designed to limit the number of free parameters such that any experimental results suggest straightforward answers. This unsatisfactory situation is slowly being replaced with improved technology such as FRET, which can record concentrations of compounds in single cells in real time.

## 2. Techniques

As this review will focus primarily on lactate, we will discuss only those techniques used to measure this compound. Lactate comprises three carbon atoms and is a charged molecule at physiological pH [9]. The lactate content of any tissue can be easily measured ex vivo by using biochemical assay and accurately quantified when compared to standard calibration. Whilst this can give accurate estimations of gross lactate levels at relatively low resolutions, i.e., using large volumes of tissue, it does not provide any impression of lactate concentrations changes over time.

### 2.1. Dialysis

This technique uses probes inserted into regions of the brain with the extracellular fluid being sampled regularly to allow post hoc measurements of lactate in live animals via biochemical assay [10]. Such measurements do allow the assessment of lactate longitudinally but not in real time. The accuracy of the technique is questionable, as the insertion of the probe damages surrounding tissue resulting in the release of intracellular contents into the extracellular fluid. Given the presumed high concentrations of intracellular lactate, this could result in higher estimates of lactate. The probe itself will damage tissue such that localised function at the dialysis tip may be compromised.

### 2.2. Biosensors

These sensors tend to be platinum wires coated with an enzyme that responds to lactate, e.g., lactate oxidase, which creates an amperometic reaction, that can be used to estimate lactate levels based on concentration dependent calibrations [11]. The diameter and length of the sensors vary but are generally suited to recording from large areas of tissue, with lengths of 5 mm and 50 μm in diameter being typical, although smaller variations are available. Since these electrodes are placed outside the tissue, it must be appreciated that a stable recording of lactate levels means there is a steady efflux of lactate from the tissue, i.e., from the intracellular to the extracellular compartments. Any decrease in the lactate signal denotes a decrease in the rate of flux, which may be either decreased output or increased uptake of lactate, presumably into the neuronal compartments. These biosensors offer the advantage of allowing real-time estimates of lactate with qualitative assessment of lactate changes. The sensors can also be implanted into the brain of live, awake animals [12,13,14]. Such recordings are extremely useful in measuring the changes in lactate in response to experimental manoeuvres, e.g., evoked activity. The sensors have a limited life, usually a few days to a week, and must be calibrated at the start and end of the experiment to account for decreases in sensitivity. Given the generally large concentrations of lactate that are present in brain tissue, the signal is sufficiently large to preclude the complication of subtraction of a null sensor [15].

### 2.3. FRET

The Forster Resonance Energy Transfer (FRET) system uses nano-sensors that are sensitive to a specific metabolic compound of interest, e.g., glucose, lactate. Details of the technique can be found elsewhere [16]. Briefly, FRET sensors are fusion proteins containing a ligand-binding site coupled with two fluorescent proteins. The binding of the ligand causes changes in the orientation of the fluorescence proteins that can be measured and calibrated to estimate [ligand]. The FRET sensors can be expressed in identified cells using transgenesis or viral transduction. The nano-sensors are embedded in cells allowing for the first time the estimation of the concentration of metabolically relevant compounds within the physiological concentration range in real time [17]. Such technology has revealed intracellular pools of lactate within astrocytes in excess of 1 mM, which act as reservoirs for ready release into the extracellular fluid via lactate specific channels [18], evidence of a metabolic coupling that exists between astrocyte supply of lactate and neuronal activity [19].

### 2.4. Spectroscopy

This technology uses Magnetic Resonance Imaging (MRI) technology to allow the estimation of lactate in live subjects, be they animal models such as rodents [20], or humans [21]. This technique uses a radio-labelled carbon incorporated into the substrate of interest, e.g., glucose, to follow its metabolic fate [22]. Spectral analysis is used to estimate the presence and location of the metabolites of the labelled compound. The limitations include access to a magnet along with the considerable associated costs. In addition, there is a compromise between the volume of brain imaged and the resolution [23], and the time taken to acquire these signals means that it is not possible to measure in real time, but snap shots of the signal are regularly measured, e.g., every 10 min. This technique is valuable for assessing control human subjects versus test subjects (e.g., disease conditions—see later) to assess whether there are any gross differences in the concentration of metabolites of interest in a defined region of the brain.

## 3. Lactate in the Extracellular Fluid

The data acquired using the techniques described above reported significant lactate present in the extracellular fluid in the brain [24]. This lactate was initially assumed to be a waste product as a result of incomplete oxidative metabolism as predicted by the CMR of less than six. As such, it could be viewed as similar to the lactate that accumulates in muscle during exercise when the level of fitness of the individual, VO_2_ maximum, dictates the threshold when oxygen uptake cannot match glucose usage and lactate inevitably accumulates [25]., Studies have indicated that skeletal muscle derived lactate can serve the useful function of fuelling the brain, sparing glucose for use by muscles [26,27,28]. A similar reassessment of the role of brain lactate present in the extracellular fluid asserts that it should be viewed as a “substrate in transit” rather than as a waste product. It has been known for decades that the following broad generalisation can be applied to neurones and glial cells in the mammalian brain, where neurones are primarily oxidative and glial cells (astrocytes) are primarily glycolytic and are net lactate producers [29,30,31,32,33].

### 3.1. Lactate Efflux from Astrocytes

The discovery of lactate efflux from astrocytes came from tissue culture studies in which an exclusively astrocytic population of cells could be cultured in a dish. Such studies did unequivocally demonstrate that astrocytes consume glucose and release lactate [34,35], but the experimental conditions are removed from in vivo conditions, where astrocytes are present with neurones and there is a very narrow extracellular space, which tends to concentrate any lactate released. Comparable studies using neurones in culture have demonstrated a lactate uptake but again the conditions were artificial [36]. In studies where neurones and astrocytes were co-cultured, there emerged the first evidence of an active role for lactate, and by extension, a functional role for astrocytes in the metabolic support of neurones. Hypothalamic neurones were more likely to survive if co-cultured in the presence of astrocytes; although, no role was ascribed to the astrocytes [37]. This was unravelled in co-culture studies in which it was the presence of glycogen in astrocytes that was the determining factor, rather than the astrocytes themselves [38]. Such a finding requires a brief digression on the role of glycogen.

### 3.2. Brain Glycogen

Glycogen is a multi-branched macromolecule consisting of a protein skeleton to which are attached dehydrated glucose (glucosyl) molecules [39]. The branching structure of glycogen ensures that numerous glucose molecules can be attached or released from the molecule simultaneously, clearly a functional advantage over a single unbranched structure, which would offer very limited scope for turnover [39]. The main depots of glycogen are in the liver and skeletal muscle with liver having the highest concentration but skeletal muscle having the largest amount [39]. Liver glycogen acts as a central store from which glucose can be released directly into the systemic circulation in order to sustain normoglycaemia and sufficient delivery of glucose to the brain [40]. In skeletal muscle, glycogen is metabolised glycolytically to lactate to fuel muscle function. This process is inefficient and yields relatively little ATP with the excess lactate produced contributing to the lactate threshold that is responsible for muscle exhaustion [41], although recent evidence points to this lactate fuelling brain function [26]. Glycogen is synthesised by the actions of glycogen synthase and metabolised by glycogen phosphorylase to form glucose-6-phosphate, which is glycolytically metabolised [40]. An intriguing aspect of glycogen is that it is located in the brain but at relatively modest concentrations; thus, an analogous role as an energy reserve as occurs in the liver is unlikely. The role of brain glycogen is intriguing as the liver acts to supplement glucose delivery to the brain when systemic levels of glucose fall so brain glycogen must have a specialised role that cannot be met by systemic glucose and may act as a short-term energy buffer for specialised functions [42].

### 3.3. Lactate Shuttling between Astrocytes and Neurones

The first reports of roles for brain glycogen emerged in 1994 from studies on honeybee retina, a convenient model to study metabolism given its metabolic and morphological compartmentalisation [43]. This was the first study to demonstrate metabolic cell-to-cell signalling in which glial cells played a key role. In response to stimulation, the glial cells take up glucose, which is converted to alanine. The alanine is then transported out of the glial cells into the extracellular fluid from where it is taken up by the neuronal elements, the photoreceptors, for oxidative metabolism. During this process, there is an increase in oxygen uptake in the photoreceptors but not in the glial cells [43]. This paper was extremely important as it confirmed a metabolic function for glial cells that had been proposed a century earlier by Cajal. Based purely on their location between blood vessels and neurones, where the neurones made no direct connection with blood vessels, he proposed astrocytes play a role in supplying nutrient to neurones [44]. The honeybee paper validated the idea of reduced models to imply roles in more complex mammalian systems. The study upset the neuro-centric mindset that prevailed in neuroscience at the time, as it was indicative of a prime role for astrocytes in nervous system metabolism. It suggested that malfunctions in astrocytes could compromise the integrity of neuronal energy requirements and suggested a complex chain of requirements concerning the location of metabolic enzymes and substrate transporters. The astrocyte neurone lactate shuttle hypothesis (ANLSH) proposed in a contemporaneous paper is a similar scheme for the mammalian brain, with the exception that lactate was the conduit between astrocytes and neurones, and the glucose uptake was coupled with the glutamate uptake into astrocytes that is a consequence in of increased synaptic activity [45].

In the last 25 years, the role of lactate as a conduit between astrocytes and neural elements has flourished and is associated with both physiological functions and pathology associated with several distinct disease processes. It is timely to question why such a metabolic pathway evolved and to assess the benefits and drawbacks. The dogmatic view of direct glucose uptake by neurones is consistent with the presence of glucose transporters on neuronal membranes [46]. The millimolar concentrations of glucose in the extracellular fluid present a readily accessible pool of glucose for short-term requirements. However, any longer-term requirements would be met via increased delivery of systemic glucose, which is not a rapid process, being mediated via activation of metabotropic astrocyte receptors leading to vasodilation mediated via release of Ca^2+^ from intracellular stores [47]. Neuronal metabolism of glucose would require ATP under conditions where the balance between demand and supply is shifting to an energy deficit. The manufacture of lactate in astrocytes shifts the metabolic burden to the astrocyte thereby sparing neuronal ATP. The uptake of lactate into neurones is based on concentration gradients requiring no immediate ATP [48]. Whereas glucose requires two molecules of ATP for its glycolytic metabolism, lactate requires no such upfront ATP and can be oxidatively metabolised [2]. Another aspect is speed, where the production of lactate in astrocytes from glycogen is a rapid process and produces a readily available pool of lactate [18]., It must be appreciated that there is supporting evidence that glycogen metabolism in astrocytes is used purely to fuel astrocytes and that neurones primarily use glucose, not astrocyte derived lactate, for their energy requirements.

### 3.4. Lactate Shuttling in the Optic Nerve

The first detailed study of lactate shuttling from astrocytes to neural elements was carried out in the rodent optic nerve, an advantageous preparation in such studies as it was a simple structure devoid of the complications of neuronal cell bodies and synapses, comprising axons and glial cells. Its cylindrical shape made it accessible to recording of the stimulus evoked compound action potential (CAP) via suction electrodes, which can be used as a real-time monitor of the number of conducting axons [49]. Simultaneous lactate measurements at the bath/nerve interface produces a real-time estimate of lactate flux from the tissue, although it must be appreciated that the released lactate would be instantly diluted by flowing bath solution [50]. In this preparation, astrocyte glycogen is metabolised to lactate which is transported to the axons when exogenous glucose is withdrawn [51,52]. Although aglycaemia is un-physiological such experiments are valuable as they indicated that lactate shuttling occurred in the optic nerve. Under these conditions, glycogen phosphorylase was inhibited by 1,4-dideoxy-1,4-imino-d-arabinitol (DAB) or isofagomine [53,54], interruption of the flow of lactate caused the loss of the CAP [42], with immuno-histochemical studies confirming the presence of the monocarboxylate transporters (MCT) required for influx and efflux of the lactate [55] (Figure 1).

It was also shown that glycogen derived lactate supported the CAP under physiological conditions when the tissue demand was increased by imposing high intensity stimulus [42,54]. This was an important finding as it suggested a physiological role where lactate acts as a supplemental energy substrate when normoglycaemic concentrations of glucose were insufficient to support conduction. The CAP recorded during 100 Hz stimulus was shown to be fully supported when the nerve was bathed in 30 mM glucose, but failed when 10 mM glucose was supplied, indicating that there were no absolute thresholds with regard to CAP failure, but that the CAP could be maintained as long as the energy supply exceeded the demand [51]. The summary of this set of experiments was that lactate shuttling existed in the rodent optic nerve under both pathological and physiological conditions. Lactate sensors revealed that there is a steady efflux of lactate from the tissue, probably from astrocytes, but the flux fell rapidly to zero on introduction of aglycaemia, indicative of the energy requiring axons taking up all available lactate reflected in the rapid fall in lactate measured by the sensor [56].

## 4. Physiological Roles for Lactate Shutting

### 4.1. Learning and Memory

Studies in rodent hippocampus have revealed the importance of lactate shuttling during learning and memory. The initial comprehensive study used both electrophysiology to study long-term potentiation (LTP) in hippocampal slices and whole animal behavioural work [57]. The slice studies showed that LTP could be demonstrated in CA1 neurones as a result of Schaffer collateral stimulation with the increased fEPSP slope, the classic indication and monitor of increased synaptic efficacy. Application of the glycogen phosphorylase inhibitor DAB prevented the maintenance of LTP but this effect could be circumvented by exogenous application of lactate, indicating that lactate uptake by neurones was required for the induction of LTP and the neurones must have access to the lactate during the conditioning phase. Application of lactate after the conditioning stimulus does not restore LTP, suggesting that the conduit role of lactate as a transportable energy substrate in LTP was analogous to the optic nerve studies, where the energy-dependent conditioning stimulus imposes a heavy metabolic demand on the tissue. If there is no access to lactate at this time, LTP does not occur, indicating that when tissue energy demand exceeds supply there is a functional failure. The behavioural tests were conducted using an aversive fear conditioning protocol as a test for memory. Rats were placed in a chamber connected to another chamber, which was accessed via a door. The natural inquisitiveness of the rats compels it to explore and it will soon enter the second chamber. When this occurred, the rat was given an electric foot shock and then replaced in the initial chamber. The time taken for the rat to re-enter the second chamber is an indication of its memory, the longer the latency the greater the memory. The latency under control conditions was about 300 s, but this was greatly reduced when the rat was injected with DAB 15 min before the conditioning pulse, and the inhibitory effect of DAB lasted for up to a week. Injection of DAB after the conditioning pulse had no effect on the latency. Sampling of extracellular fluid lactate with a dialysis probe in the hippocampus showed elevations in lactate in the immediate aftermath of the conditioning pulse that relaxed back towards baseline after about 30 min, but this lactate increase was abolished when DAB was injected into the hippocampus prior to the conditioning stimulus. The study demonstrated for the first time a role for lactate shuttling between glia and neurones in a physiological process, i.e., learning and memory [57] (Figure 2). The importance of this is reflected in the studies that followed in its wake.

This study described above made an extremely interesting point, which was that lactate uptake was required by neurones to meet the energy demand of LTP induction even in the presence of normoglycaemic concentrations of glucose. Under physiological conditions the source of this lactate was astrocytic glycogen but experimental interventions indicated that exogenously applied lactate could fulfil this role in the event of glycogen metabolism block [57]. This was followed up in a subsequent study in which the role of alternate substrates and transport proteins were investigated. Using a similar experimental protocol as described above injection of DAB into the dorsal hippocampus reduced the latency of rat entry into the shock compartment post conditioning, but this latency could be increased towards control levels by the co-injection of pyruvate or β-hydroxybutyrate (β-HB) with the DAB, but not glucose [58]. Expression knockdown was performed on the MCT1 and MCT4 transporters, which are expressed on astrocyte membranes. Both knockdown models reduced the latency post-conditioning, but this could be counteracted by the introduction of pyruvate or β-HB. Knock out of the MCT2 transporter, which is expressed exclusively in neuronal membranes, caused a decrease in latency that was not affected by the addition of pyruvate, a result expected if lactate or pyruvate uptake into neurones was an integral part of the pathway. Further experiments indicated that lactate induced protein synthesis with translation rates estimated using the Surface Sensing of Translation (SUnSET) technique. This technique involves the use of an anti-puromycin antibody for immunological detection of puromycin-labelled peptides, and allows for the detection of changes in protein synthesis. Learning leads to increases in translation of mRNA expressed in excitatory neurones, but the effect on inhibitory neurones was not known. The study demonstrated that training increased mRNA translation in both excitatory and inhibitory neurones and that inhibiting glycogenolysis blocked these increases, which could be restored by co-injection of DAB with L-lactate. The mechanism by which lactate increases translation was shown to be via the Arc/Arg3.1 cytoskeleton associated protein in the CA1 region of the hippocampus [58].

Another study focussing on learning and memory revealed the differences between learning and memory retrieval [59]. The compound dichloroacetate (DCA) inhibits pyruvate dehydrogenase (PDH) kinase, enhancing the activity of PDH and allowing a higher rate of transfer of pyruvate to the mitochondria, thereby reducing cytoplasmic pyruvate and reducing its reduction to lactate. DCA was injected into the frontal cortex and hippocampus of rats. In tasks that model spatial learning, a process that requires communication between the hippocampus and the frontal cortex, DCA injected 30 min before the training paradigm resulted in a significant reduction in learning compared to saline treated animals. DCA administration did not interfere with memory retrieval, which was modelled by the Morris water maze after 4 days of training, where there was no difference between treated and control mice. The results suggest that anaerobic glycolysis, i.e., lactate production, is required for memory acquisition, but not for retrieval [59].

Using knock down expression of the astrocytic MCT4 and neuronal MCT2 transporters in models of depression, a requirement for lactate shutting for spatial information acquisition and retention in the hippocampus dependent tasks was demonstrated. Intra-cerebral infusion of lactate resulted in mice with reduced MCT4 expression improved memory but had no effect in mice with MCT2 knock down, consistent with astrocytic release of lactate followed by neuronal uptake in neurones. However, only MCT2 is required for long-term (7 day) memory formation. The results are consistent with the role for lactate as an energy substrate for the high-energy demand of de novo mRNA translation requited for long-term memory consolidation [60].

The role of lactate in supporting expression of genes involved in plasticity was investigated in cultured neurones from mouse neocortex. Lactate potentiates NMDA mediated currents causing elevated intracellular Ca^2+^ via increased Ca^2+^ influx. Lactate also increases NADH, altering the redox state of the neurones, and NADH mimics the effects of lactate on NMDA, therefore it is likely that the NMDA receptor is the target for NADH. The genes that are upregulated include Arc, c-fos and Zif268. Lactate is specific for these effects as neither glucose nor pyruvate reproduces lactate’s ability to modify these genes. The lactate effect occurs over a short period of time and both levels of mRNA and proteins are increased [61].

In a mouse model, lactate was demonstrated to rescue memory in animals treated with DAB in a similar way to that previously demonstrated in the rat [57]. 3-D morphological reconstructions using transmission electron microscopy in mice injected with DAB demonstrated there were fewer synaptic spines per unit volume than control mice, but the effect could be reversed by co-injection with lactate and DAB. LTP in control mice produces new spines at 24 h post-conditioning treatment, but this effect is reduced by DAB. Co-application with lactate restored the appearance of spines. Lactate did not rescue spines or increase the postsynaptic density. Spine density increased after learning but DAB prevented this increase, whereas DAB with lactate co-injected rescued the memory. DAB inhibited spine size and postsynaptic density surface area, but this was not rescued by lactate. After learning glycogen granules grew bigger, but this was blocked by DAB. Glycogen granule clustering increased with learning but this decreased with DAB and was not rescued with lactate [62].

In awake, freely behaving rats, LTP was induced using high frequency stimulus. Lactate sensors implanted in the dentate gyrus showed delayed elevations in lactate with stimulus associated with a brief transient drop in lactate followed by a larger increase, which relaxed towards baseline. This result mimics those shown in rats where stimulus resulted in a transient decrease in lactate followed by a sustained elevation [24]. The increase in lactate was potentiated starting 24 h after LTP induction. The lactate increases showed circadian rhythmicity being highest during the dark period, with these alterations in lactate levels potentially reflecting a long-term change in a metaplastic effect [63].

Complementary information in chicks, which commenced in the 1990s, has demonstrated the requirement for glycogen metabolism in learning and memory and can be summarised as follows. Glycogen levels decreased in the forebrain of the chicks immediately following the conditioning paradigm, consolidation of the memory was attenuated by treatment with DAB, whose effects could be circumvented by the addition of acetate, which is metabolised in the astrocytes [64,65,66].

### 4.2. Sleep

Implanted lactate sensors in rats show a decrease in lactate during slow wave sleep compared to the spontaneously waking state, and lactate was lower than waking during paradoxical sleep, but during active waking lactate rose significantly. There was a circadian aspect associated with the lactate levels, with the lowest level occurring during light and the highest during the dark period. It must of course be borne in mind that rodents are nocturnal, being active during the dark period. The relative levels of lactate during waking, slow wave sleep and paradoxical sleeping were maintained in light and dark, but the background levels were increased in the dark phase compared to light. Historical studies show glucose uptake is lower in slow wave sleep. Data suggest that during waking and paradoxical sleep there is an increased availability of glucose and hence an increased availability of lactate to fuel function. The lactate levels measured suggest a balance between lactate production and neuronal uptake. The decrease in lactate during slow wave sleep represents the quiescent state and suggests it is restorative when glucose levels in astrocytes are increased. During transition from waking to paradoxical sleep, there is increased interaction between neurones and astrocytes and hence the lactate levels increase since it is produced in astrocytes and moved to neurones [67].

During sleep deprivation, lactate dehydrogenase (LDH), which converts pyruvate to lactate, is increased, as is the Glut1 transporter protein, which facilitates glucose uptake into astrocytes and increases lactate production, suggesting this upregulates the lactate shuttle during wakefulness [68].

Glycogen is known to accumulate in the brain during sleep and decreases during wakefulness. It is postulated that insufficient glycogen breakdown during sleep deprivation leads to elevated K^+^ as the decreased availability of glycogen-derived energy substrate in the astrocyte ensures inadequate astrocytic uptake of interstitial K^+^, leading to impairment of interstitial K^+^ buffering, which is an energy dependent process. A fluorescent K^+^ indicator was used to detect changes in extracellular K^+^. DAB was used to inhibit glycogen breakdown, which resulted in extracellular K^+^ rises and activation of neuronal pannexin 1 channels and downstream inflammatory pathways, which may cause migraines [69].

### 4.3. Decision Making

The anterior cingulate cortex (ACC) is a component of the functional circuit involved in mediating perception and processing chronic visceral pain. There is impairment of LTP in ACC neurones of rats with visceral hypersensitivity (VH) and this implies that pain associated synaptic metaplasticity may lead to emotional and cognitive disturbances. In the in vivo model of VH, application of exogenous lactate improved the decision-making process in the rat gambling test. Lactate rescued the faulty ACC LTP and saved DAB induced impairments of ACC spike field synchrony. It was shown that under normal physiological conditions release of lactate from astrocytes plays a role in LTP and during VH this lactate release is reduced. The cause of the reduction in lactate release was astrogliosis, in which the astrocytes respond to the VH by forming glial scars, altering basic astrocyte function where the EAAT transporter that facilitates glutamate uptake into the astroglia is down regulated. Optogenetic activation of astrocytes in the ACC caused release of lactate. Experiments on awake rats showed optogenetic stimulation of astrocytes in the ACC rescued the decision making process and allowed synchronisation of the basolateral amygdala to ACC neurones [70].

To illustrate the range of different physiological roles for lactate shuttling in both the central (CNS) and peripheral nervous systems (PNS), it is evident that lactate transfer between astrocytes and neurones is involved in myelination where MCT1 expression is required on Schwann cells for proper myelination to occur in sensory but not motor neurones, resulting from altering metabolism in glycolysis and mitochondria [71]. A similar study showed that known down of the MCT1 on Schwann cells disrupted neuromuscular innervation [72].

These results illustrate that certain physiological processes that occur in the nervous system rely on the transfer of lactate from astrocytes to neurones, suggesting this is a common mechanism that is present in both the CNS and PNS, in which neurones display a metabolic versatility with regard to substrate use. The presence of an extracellular pool of lactate that acts as a reservoir of readily available substrate is clearly advantageous to neurones that require a rapid uptake of energy substrate as a result of instantaneous increase in tissue energy demand as occurs during LTP, where the conditioning pulse paradigm is 2 × 100 Hz stimulus for 1 s separated by 10 s. The rate at which the pool of lactate can be refilled is clearly of prime importance and highlights a role for glycogen since glycogen phosphorylase is a rapid enzyme that is able to metabolise glycogen to lactate faster than the glucose to lactate pathway, since it does not require phosphorylation of glucose. The lactate present in astrocytes has ready rapid access to the extracellular pool via the lactate channels [18], which do not rely on the balance between lactate and H^+^ gradients that the MCTs rely on [48]. The disadvantages of such a scheme include that the neural elements are reliant upon astrocytes for lactate delivery, i.e., the more links in the chain the more susceptible it is to disruption. The areas of weakness can be identified as glucose delivery to astrocytes, glucose uptake into astrocytes via Gluts, metabolism to lactate, lactate transport out of astrocytes via MCTs and the lactate channel, lactate uptake into neurones via MCTs and efficient neuronal metabolism of lactate. The vulnerability of the chain has been associated with a variety of pathological conditions.

## 5. Pathological Conditions

### 5.1. Alzheimer’s Disease (AD)

Studies in female mice of both control and AD models, which expressed amyloid and tau pathologies as well as synaptic dysfunction (TgAD), revealed that in the control mice there was a decrease in general glucose uptake at an early stage of ageing in the normal brain which reached significance at 6 to 9 months, a deficit that was sustained at 12 to 15 months. Immunohistochemical assessment of expression of a variety of neuronal and astrocytic glucose transporters demonstrated a decrease in neuronal Glut3, which was correlated with the decrease in glucose uptake. In the AD model, there was a qualitatively similar decrease in glucose uptake, but there was a decrease in blood brain barrier Glut1 expression and neuronal Glut3 expression, which were positively correlated with decreased glucose uptake. The level of hexokinase protein expression and activity was decreased in the control mice, which positively correlated with decreased glucose uptake, but the pyruvate dehydrogenase (PDH), which converts pyruvate to lactate and links glycolysis with mitochondria, was increased. In the AD model, there was also increased phosphorylated PDH in the hippocampus, which is indicative of decreased PDH activity. LDH interconverts lactate and pyruvate with the astrocytic 5 isoform converting pyruvate to lactate and neuronal 1 subtype converting lactate to pyruvate. In control mice, there was a decrease in expression of both these enzymes indicating a deficit in lactate production and use. Investigations of MCT expression in control mice showed decreased astrocytic 4 expression but the blood brain barrier and astrocytic 1 expression were not affected, and there was a downward trend in the neuronal 2 expression. β-hydroxybutyrate, a ketone body, was increased at 12 h, indicative of an alternate source of energy substrate. These results suggest a sequence of events associated with ageing that comprise glucose hypometabolism and mitochondrial dysfunction, which leads to a bioenergetic phenotype of people at risk from AD [73].

These results were supported by a study in which there was a decrease in lactate in the hippocampus and the cortex of rats in the Aβ_25-35_ model of AD compared with control rats. In addition, their performance in the Morris water maze was worse and the MCT2 transporter, which is expressed in neurones, was decreased. This indicated that lactate production and MCT2 expression are down regulated in AD suggesting insufficient neuronal energy metabolism contributes to the disease (Figure 3).

A separate study confirmed this data in an AD model mouse (APP/PS1), which displays amyloid plaques and demonstrated a general decrease in lactate content in the brain, down regulation of MCT1, 2 and 4 transporters and a decrease in LDHB in neurones, which is involved in conversion of lactate to pyruvate [74].

Thus, AD appears to be consistent with both decreased lactate availability for neurones and decreased expression of MCTs to transport lactate from astrocytes to neurones. The enzymes associated with the pathway are down regulated, which speaks of an overall decrease in lactate use by neurones and decreased glucose uptake, suggesting neuronal energy metabolism is compromised in AD [75]. Additional supporting evidence for a role in lactate shuttling can be found in the following references [76,77,78,79].

### 5.2. Amyotrophic Lateral Sclerosis (ALS)

ALS is a disease not usually associated with the metabolic co-operation between astrocytes and neurones. In SOD1 transgenic mice, an established model of ALS, there was a decrease in the lactate content of spinal cord homogenates. In addition, transcriptome analysis revealed a decreased expression of enzymes involved in carbohydrate metabolism as well as down regulation of the astrocyte lactate transporter and the Glut1 glutamate transporter. Co-culturing SOD1 astrocytes with motor neurones leads to a decreased neuronal viability accompanied by about a 30% decrease in lactate release from these cells. This deficiently can be circumvented if exogenous lactate is added. These data are suggestive of an interaction between astrocytes and neurones in the pathology of ALS. Culture studies are convenient to gauge the interactions between the cell types, but may be too simple/reduced to fully account for the in vivo disease process. There is down regulation of the lactate transporter and a lower concentration of lactate is recorded. It was shown that it was crucial for there to be a neuronal interaction with the SOD1 astrocytes to reduce lactate release [80]. A summary of the role of energy metabolism in ALS has recently been published [81].

### 5.3. Depression

There is increasing evidence of an involvement, and, in particular, of a metabolic involvement, of astrocytes and metabolic coupling to neurones in depression. Experiments conducted in mice undergoing behavioural testing similar to those described earlier in this review were treated with IV injection of lactate, which was shown to result in elevated lactate in the hippocampus. Injection of lactate decreases the time the mice spent immobile compared with un-injected mice in the forced swim test. Lactate down regulates the relative levels of phosphorylation of pGSK3a, b and CREB, which are targets for antidepressant drugs such as lithium. Lactate upregulates Arc mRNA in the hippocampus, and lactate injection also causes a decrease in COX expression, which is responsible for the production of prostaglandins. Lactate injection affected genes associated with 5-HT including regulation of the 5-HT receptor, p11, S100β an astrocyte marker and the transcription factor He5. The data can be summarised as lactate release from astrocytes acts to regulate the expression of proteins involved with 5-HT receptor trafficking [82]. A summary of the role of energy metabolism in depression has recently been published [83].

There is confounding information that indicates there is excess glucose and lactate in the brain during depression in a study that used H-MRS to estimate brain levels of the substrates in control and sufferers of depression in the anterior cingulate in humans. There appeared to be a linear correlation between severity of depression and glucose levels in the brain. However, the limited sample size and the limited resolution of the techniques require fine-tuning [84].

### 5.4. Schizophrenia

The DISC gene is a risk factor for schizophrenia. Transgenic mice with reduced DISC had reduced levels of Glut 4 and glucose uptake into astrocytes. This was associated with reduced glycolysis and oxidative phosphorylation and decreased lactate production. Exogenous application of lactate rescued the abnormal behaviour of DISC transgenic mice. The results suggest that malfunction of the shuttling of lactate to neurones is involved in the behavioural phenotype associated with mutation of this gene [85]. There is increasing evidence of a bioenergetic component to schizophrenia in which there is disrupted metabolic glia to neurone lactate signalling. Analysis of post mortem brain dorsolateral prefrontal cortex in subjects with schizophrenia, and two mouse model of schizophrenia, found increased lactate in the brain, where there was a decrease in lactate in the DISC mouse [86].

### 5.5. Stress

Stress in mice, which was induced by a forced 20 min swim, induced altered genes associated with connexons and cell growth and caused astrocytic hypertrophy and prolonged Ca^2+^ elevations in astrocytes. Stress also decreases astrocytic gap junction channel expression and impairs functional coupling. LTP depends upon the integrity of the astrocyte syncytium and in mice with the stress phenotype LTP could not induced. Blocking the continuity the astrocyte syncytium inhibits LTP, but this blockade can be circumvented by adding exogenous lactate [87].

Stress elicits the release of glucocorticoids (GC) that affect energetics. Astrocytes express GC receptors, and elimination of these receptors in astrocytes resulted in animals that presented with impaired aversive memory expression in studies using contextual fear conditioning and conditional place aversion. Genes associated with glucose metabolism were adversely affected and the stress response molecule SGK1, which is involved in glucose uptake, undergoes GC medicated modulation and was down regulated in astrocytes, but not in neurones [88]. Short-term stress improves memory but long term decreases it in a mouse model of learning and memory where β1 arrestin controlled levels of MCT4 on astrocytes and LDHA, which improves memory function [89].

## 6. Regeneration

### 6.1. CNS Regeneration

A role for shuttling of lactate between glial cells and axons has been proposed in the regeneration of injured axon in both the central (CNS) and peripheral nervous systems (PNS). In the CNS, there is active inhibition of regeneration by the action of myelin associated proteins and release of CSPG by astrocytes forming the glial scar [90], thus manipulating this inhibitory environment is viewed as being a viable therapeutic manoeuvre to aid regeneration. In a study, both drosophila and mouse models of injury to CNS axons were used. In drosophila, the glial cells were reprogrammed to activate PI3K, EGFr and cell cycle regulation pathways and showed a change from an inhibitory environment to one that encouraged axon regeneration. The effect was mediated by a change in the metabolic profile of glial cell, which switched to a glycolytic phenotype and produced excess lactate, which acted on GABA_B_ metabotropic receptors, which were present on the neurones and promoted elevations in intracellular cAMP. In the mouse model of spinal cord injury, lactate encouraged axon regeneration and improved functional scores of tests designed to monitor motor improvement, for the first time providing conclusive evidence for a role for lactate shuttling in regeneration and identifying a clinically relevant therapeutic target [91].

### 6.2. PNS Regeneration

The peripheral nervous system differs from the CNS in that axons are capable of regenerating post-injury in a manner in which central axons are not, where there are none of the inhibitory processes present in the CNS. The energy requirements of PNS regeneration were investigated in a study in which the sciatic nerve was crushed, and the regeneration of the axons was monitored both electrophysiologically and morphologically. In mice in which the MCT1 transporter expression was reduced, this transporter is located on perineurial cells, DRG neurones and Schwann cells, thereby providing shuttling lactate to the regenerating axons. There was delayed recovery or failure of axon regeneration in both motor axons as assessed by myography, and in sensory neurones estimated morphologically in the transgenic mice suggesting an active role for lactate shuttling during PNS regeneration [92].

The role of Schwann cell lactate shuttling in PNS axonal regeneration was further investigated in a study in which peripheral nerves were injured in a mouse model. In response to the energy requirements associated with regenerating axons, the Schwann cells underwent a phenotypic change in which they increased their glycolytic capacity by increasing the expression of all glycolytic enzymes. This increased the production of pyruvate and lactate and increased expression of the glucose transporter Glut1 on Schwann cells, facilitating glucose uptake. LDHA was increased which converts pyruvate to lactate, and was likely involved in the reported elevated uptake and metabolism of glucose and increased production and release of lactate. Soon after axotomy there was increased expression of MCT1 and MCT4 transporters, which serve to facilitate lactate exit from the Schwann cell. The rapid increase in AMPK activated protein kinase and mTOR complex 1/2 in the aftermath of injury is suggestive that these signalling pathway are associated with the lactate release and uptake by the axons [90] (Figure 4).

## 7. Conclusions

In the last decade, our understanding of the role of lactate shuttling in the nervous system has increased significantly. The transfer of lactate produced in astrocytes to neurones via the extracellular fluid was initially shown to occur during periods of limited glucose availability, but has now been expanded to include such key brain functions as learning and memory, sleep and plasticity. The multiple stages involved in such shuttling render it vulnerable to interruptions and deficits in shuttling are associated with such disparate disease states as Alzheimer’s disease, amyotrophic lateral sclerosis, stress, depression and schizophrenia. Although this area of research is in its infancy, targeting lactate shuttling to neurones may be a therapeutically viable target in a variety of disease conditions.

## Figures and Tables

**Figure 1 biosensors-10-00183-f001:**
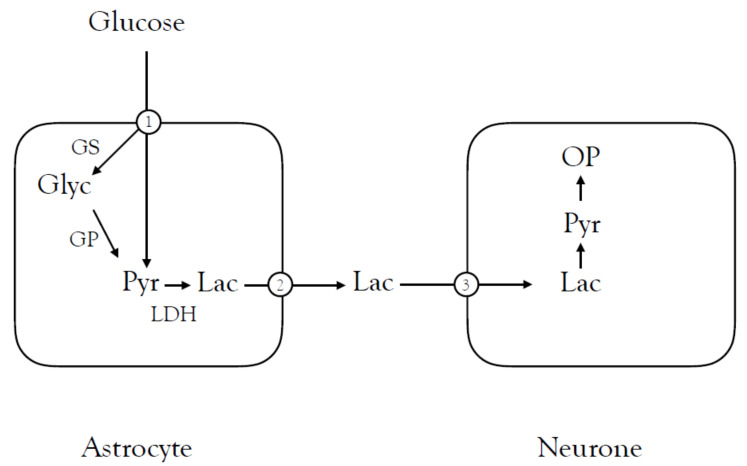
Lactate shuttling between astrocytes and neural elements in the central nervous system (CNS). Glucose enters the astrocyte via the Glut1 transporter (1) and is either glycolytically metabolised to pyruvate or stored as glycogen via the action of glycogen synthase (GS). The glycogen (glyc) is metabolised to pyruvate by glycogen phosphorylase (GP). Pyruvate is converted to lactate by lactate dehydrogenase (LDH) and leaves the astrocyte via the MCT1 transporter (2) and is taken up into the neural element via the MCT2 (3) transporter where it is oxidatively metabolised (OP). Glycogen phosphorylase can be blocked by isofagomine and DAB and the MCT2 transporter can by blocked by cinnemate (CIN).

**Figure 2 biosensors-10-00183-f002:**
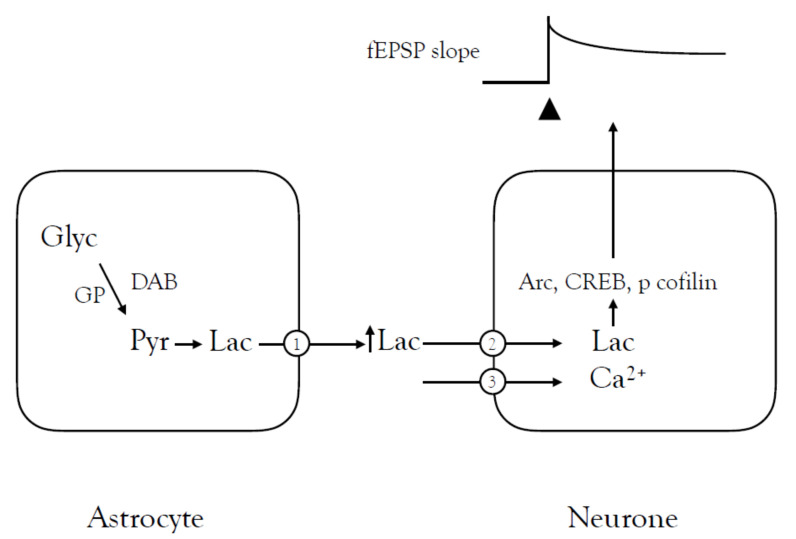
The role of lactate shuttling in learning and memory. A conditioning pulse of high frequency stimulus (100 Hz) initiates the learning process, which is reflected in the sustained increase in the field EPSP slope (upper right image). There is a transient elevation in lactate in the extracellular fluid following the conditioning pulse. The learning can be disrupted by injection of 1,4-dideoxy-1,4-imino-d-arabinitol (DAB, an inhibitor of glycogen phosphorylase) into the hippocampus or applying the N-methyl-D-aspartate (NMDA) receptor (3) antagonist MK-801. This DAB induced inhibition can be circumvented by exogenous application of lactate. Knock down expression of the MCT1 (1) and MCT2 (2) transporters attenuates the learning, but in the case of the MCT1 this can be circumvented via the exogenous application of lactate.

**Figure 3 biosensors-10-00183-f003:**
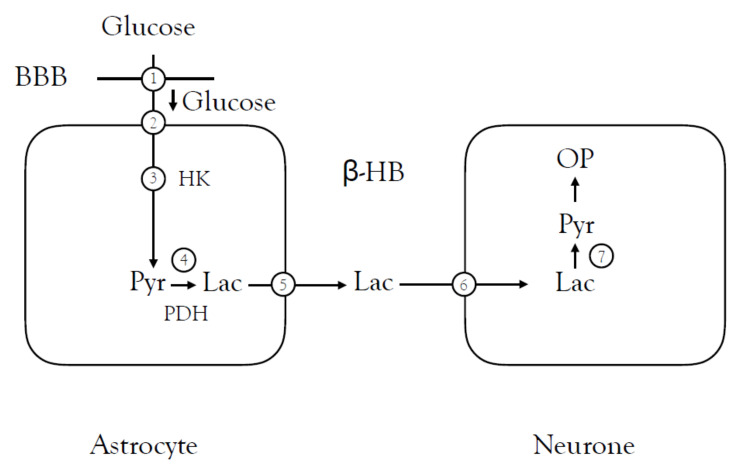
Disruptions in lactate shuttling in Alzheimer’s disease. In mouse models of AD, there is a generalised decrease in glucose uptake at 6 to 9 month of age, which was sustained at 12 to 15 months, facilitated by down-regulation of the blood brain barrier (BBB) Glut1 (1) and the neuronal Glut3 (2) transporters. The levels of hexokinase expression (3) were reduced but PDH activity (4) was increased, which reduced conversion of pyruvate to lactate. The expression of the astrocytic MCT4 (5) and the neuronal MCT2 (6) were down regulated, but the levels of the ketone body β-hydroxybutyrate (β-HB) were increased. The lactate is converted to pyruvate (7).

**Figure 4 biosensors-10-00183-f004:**
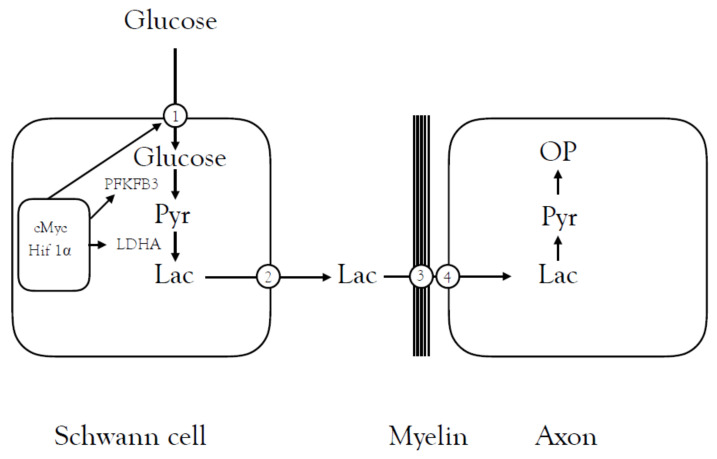
The role of lactate shuttling in axon regeneration in peripheral axons. In the aftermath of injury to peripheral axons, there is an upregulation of the transcription factor targets cMyc and Hif 1α, which leads to increased expression of the Glut1 transporter (1) facilitating glucose uptake. The glycolytic activator PFKFB3 is upregulated as is the enzyme LDHA, which converts pyruvate to lactate. The MCTs that facilitate lactate efflux from the astrocyte (2) and uptake into axons are present on the myelin (3) and axon (4) membranes.

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
