# Peer review of "Universal Glia to Neurone Lactate Transfer in the Nervous System: Physiological Functions and Pathological Consequences"

_biosensors, 2020, doi:10.3390/bios10110183_

Round 1

Reviewer 1 Report

This review does a good job of outlining the recent advances in the astrocyte-neuron field regarding lactate transfer. It was particularly nice to see some inclusion of pathology and links to physiology.

With that said, there are several areas that need further clarification or discussion:

1. In section 2.3 when introducing FRET nanosensors, further details about how they are genetically encoded and can be cell-type specific should be provided.

2. While I believe that astrocytes shuttle lactate to neurons, there are some in the field who believe that neurons primarily use glucose as their energy source.  For example, see the work by Gary Yellen. This controversy should be discussed somewhere.

3. Line 277- the SUnSET technique is mentioned, but should be described for clarification for the reader.

4. Section 4.1 nicely covers lactate and learning and memory. However, there are other studies in chicks (see work by Gibbs) that are also relevant for this section.

5. Line 334- regarding the light and dark periods, readers should be reminded the rodents are nocturnal to make this part more clear.

6. Section 4.2; last paragraph about sleep, glycogen, and K+. It is not clear how these are linked. How does insufficient glycogen breakdown lead to elevated K+? What does this have to do with lactate?

7. Section 4.3. on Irritable bowel syndrome needs work and clarification. An alternate title for this section would be beneficial, as some may view irritable bowl syndrome as a pathology. In this section, the parts about schwann cells and HCA1 receptors come out of nowhere. Are Schwann cells akin to astrocytes or oligodendrocytes?  How does the HCA1 receptor-mediated feedback loop reconcile with LTP, learning and memory and other hippocampal work?

8. Line 416- Why is a ketone body relevant? Increased 12 hours after what?

9. Section 5.2- Is there a link between NGF, p75 and lactate or metabolic enzymes?  If not, then it is not clear why these pathways are discussed other than showing that astrocyte-neuron interactions contribute to disease.

Minor points:

Line 35- The sentence about RC needs to be reworded to be clearer.

Line 234- “in vivo” electrophysiology implies whole animals, but hippocampal slices are mentioned. This should be reworded.

Lines 258 and 295- spelling-“shutting”.

Line 459- was this study about rats or mice?

Author Response

Please see detailed responses in attachment

Reviewer 2 Report

In the review entitled “Universal Glia to Neuron Lactate Transfer in the Nervous System: Physiological Functions and Pathological Consequences”, Dr. Brown and co-authors summarize a collection of data demonstrating the importance of lactate exchange between nerve and glial cells, both in physiological and pathological conditions.

The paper is of potential interest for the Readers of Biosensors.

A few points should be, however, addressed before acceptance:

  1. Page 2, line 48: this technique is not so new, as it was published in 1977; in the same page (line 69), please, note that lactate contains 3 carbon atoms and not “molecules”;
  2. Page 3, line 109: please, add “Magnetic Resonance Imaging” before MRI;
  3. Page 3, lines 125-127: the studies cited are not so recent (2004-2006) as written, especially when we consider the interest elicited by the topic in the last decade;
  4. Page 5, line 215: please, add the complete name of DAB (1,4-dideoxy-1,4-imino-d-arabinitol), and not only the acronym;
  5. Page 6, legend to Figure 1, last two lines: as described in other parts of the text, DBA is an inhibitor of glycogen phosphorylase and not of glycogen synthase;
  6. Page 7, legend to Figure 2, line 5: again, DBA is an inhibitor of glycogen phosphorylase and not of glycogen synthase;
  7. Page 7, line 285: please, write the full name of DCA; in addition, the effect of DCA should be explained a bit more: DCA (dichloroacetate) inhibits PDH kinase, and thus probably allows a higher activity of PDH (that, on the contrary, is inhibited when phosphorylated), and a higher transfer of pyruvate to mitochondria, taking it away from the cytoplasm and from its reduction to lactate;
  8. Page 9, line 348: The sentence “Glycogen is known to accumulate during sleep and decreases during wakefulness” is true for brain, but not necessarily for other organs and tissues; thus I suggest to change the sentence to: “Glycogen is known to accumulate in the brain during sleep and decreases during wakefulness”;
  9. 10, lines 409-411: The sentence “In the AD model there was also increased PDH activity in the hippocampus, where increased PDH phosphorylase is indicative of decreased PDH activity” is not clear: the activity of PDH has been recognized to be decreased in AD, and this effect can depend on a higher activity of PDH kinase (see, for example: Hoshi M et al,1996, Proc Natl Acad Sci USA 93:2719–2723);
  10. The important topic of altered metabolism in AD (paragraph 5.1) is based on 3 references only, and the discussion on ALS (paragraph 5.2), on one reference only; paragraph 5.3, although starts stating that “There is increasing evidence of an involvement…”, reports only two references; similarly, 2 references are reported for schizophrenia; it is fundamental to add other references to improve these paragraphs;
  11. Finally, figures are not cited in the main text.

Author Response

(The authors gave the same response as above.)
